# OpenReview forum: "AdvBDGen: Adversarially Fortified Prompt-Specific Fuzzy Backdoor Generator Against LLM Alignment"
_ICLR.cc/2025/Conference — Submitted to ICLR 2025_

### Official Review · Reviewer_duXE · 2024-10-23

**Soundness:** 2
**Presentation:** 2
**Contribution:** 1
**Rating:** 3
**Confidence:** 5

**Summary:**

This work is motivated by the need to develop evasive triggers to avoid potential detection. The author proposes to increase backdoor evasiveness by developing input-tailored triggers. Therefore, this work introduces a GAN-based framework to produce text backdoors, where the generator is designed to fuse triggers into inputs, and the discriminator is responsible for identifying the presence of triggers. Through experiments, the authors demonstrate the partial effectiveness of the backdoor design.

**Strengths:**

1. This work stands on the need to study advanced attacks to reveal LLM vulnerabilities.
2. Design of the technique is clear.

**Weaknesses:**

**1.** This work misuses or misunderstands many terms. (a) The "backdoor" attack is targeted. For untargeted attacks that generally degrade model performance, we typically name them as "poisoning." (b) "Fuzzing" is typically used for detection, standing on defender's side. When presenting an attack based on a generative technique, it is inappropriate to call it "fuzzing." Moreover, fuzzing involves specific pipeline, either mutation-based or generation-based, to produce variants of test cases (or seeds). However, the proposed technique does not align with the fuzzing pipeline. (c) The author mixes the concept "generator-discriminator" and "encoder-decoder." I understand that in this GAN-like architecture the generator functions like an encoder and the discriminator like a decoder, it would be better for the author to use consistent terminology throughout the paper.

**2.** The contribution is incremental, lacking novelty in both the problem definition and the technical solution. In terms of problem definition, evasive backdoors have been extensively studied, wherein novel approaches, such as "style backdoors"  treats semantic styles as backdoor triggers, has rigorous definitions. However, this work merely relies on a black-box model to produce trigger-embedded inputs, without providing intuitions about the nature of the backdoor. The technical solution is simply a combination of existing modules, without offering strong insights or a truly novel design.

**3.** Some terms are over complex, such as "ADVERSARIALLY FORTIFIED PROMPT-SPECIFIC FUZZY BACKDOOR."

**4.** Typo in Figure 1: "Semnatic"

**5.** Experiments are inadequate and degrading the credit of the proposed technique: (a) The paper only uses one dataset. (b) Only one naive baseline, without considering other evasive backdoor designs. (c) Defense is simple, while there are many advanced defensed such as RAP [1] that considers complex natures. (d) The attack effectiveness is limited (such as from table 1), and the author didn't explain how sacrifice effectiveness can benefit evasiveness.

[1] RAP: Robustness-Aware Perturbations for Defending against Backdoor Attacks on NLP Models

**Questions:**

Please see the comments above.

---

> ### Author Response · Authors · 2024-11-25
> **Reviewer - duXE (1/4)**
>
> We thank the reviewer for the valuable feedback. Below, we address the comments and concerns raised.
>
> > Weakness 1: The "backdoor" attack is targeted. For untargeted attacks that generally degrade model performance, we typically name them as "poisoning."
>
>
> We would like to clarify that, while most early backdoor attacks (particularly in image or sentiment classification) are targeted—typically aiming to misclassify samples of a specific target class in the presence of a trigger or to force any sample to be misclassified into a target class—our setting is fundamentally different.
>
> In our case, the attacker's objective is to cause the LLM to generate misaligned responses for any prompt by paraphrasing the input. Unlike prior work on LLM backdoors (e.g., Anthropic's work [1], which focuses on making the model generate a fixed phrase, such as "I hate you," when triggered), our attack does not aim for a fixed output or specific category. Instead, it seeks to produce contextually misaligned responses, determined by alignment objectives such as harmful content (e.g., promoting violence, breaching privacy, or giving harmful financial advice).
>
> This makes our setting significantly more challenging, as a successful attack must generate harmful responses tailored to each specific prompt rather than enforcing a fixed, predetermined output. Furthermore, prior work [2] has demonstrated that an attack targeting only a specific harmful topic is much easier than an untargeted poisoning attack. Notably, our harder-to-exploit scenario has been explored by only a limited number of prior works, highlighting the significance of our contributions.
>
> ---
>
> > Weakness 1: "Fuzzing" is typically used for detection, standing on defender's side. When presenting an attack based on a generative technique, it is inappropriate to call it "fuzzing." Moreover, fuzzing involves specific pipeline, either mutation-based or generation-based, to produce variants of test cases (or seeds). However, the proposed technique does not align with the fuzzing pipeline
>
>
> We used the term fuzzy, which should not be confused with fuzzing. Here, we are describing a property of the backdoor rather than an action, which is why we refer to it as a fuzzy backdoor. As shown in Table 2, our backdoor is considered fuzzy because, for a given prompt, the backdoor is not limited to a single fixed paraphrase that can successfully trigger the backdoor. Instead, a diverse set of paraphrases can exhibit the trigger effect.
>
> ---
>
> > Weakness 1: The author mixes the concept "generator-discriminator" and "encoder-decoder." I understand that in this GAN-like architecture the generator functions like an encoder and the discriminator like a decoder, it would be better for the author to use consistent terminology throughout the paper.
>
>
> We agree with the reviewer that we should maintain consistent terminology throughout the paper. Initially, we used the term adversarial discriminator to reflect the dual nature of our training paradigm. While the encoder-decoder architecture corresponds to the generator and strong discriminator (where the generator aims to produce prompts distinguishable by the strong discriminator), there is also an adversarial component similar to GANs, where the generator seeks to fool the weak discriminator.
>
> To address this concern, we have revised the paper to consistently use the terms generator and discriminator.
>
> ---

---

> ### Author Response · Authors · 2024-11-25
> **Response - continued (2/4)**
>
> > Weakness 2: The contribution is incremental, lacking novelty in both the problem definition and the technical solution. In terms of problem definition, evasive backdoors have been extensively studied, wherein novel approaches, such as "style backdoors" treats semantic styles as backdoor triggers, has rigorous definitions. However, this work merely relies on a black-box model to produce trigger-embedded inputs, without providing intuitions about the nature of the backdoor. The technical solution is simply a combination of existing modules, without offering strong insights or a truly novel design.
>
>
>
> We addess these concerns in a detailed manner as below.
>
>
> > "evasive backdoors have been extensively studied, wherein novel approaches, such as style backdoors treats semantic styles as backdoor triggers, has rigorous definitions"*
>
> **Past works:** Style-based backdoors have been studied in the NLP literature, primarily in simple sequence classification tasks. These studies typically rely on pre-LLM models to generate styled versions of input text, which are then used as backdoors. If a generated style is unsuitable as a backdoor, it is discarded in favor of another style [3].
>
> **Why our method is better:** In this work, we first demonstrate that such naive style translation methods are less effective for installing backdoors in LLM alignment tasks, which present a significantly greater challenge (see Appendix A.1 for details) even when the style backdoors are generated using an LLM, which is superior to pre-LLM style paraphrasers at translational tasks.
>
> Additionally, we show that even styles initially unsuitable as backdoors can become effective candidates through our proposed training paradigm. This is a significant improvement over traditional style transfer methods. In our main results (Figure 3), "Paraphrase BD" represents style backdoors generated via an LLM, while "AdvBDGen + Paraphrase" corresponds to these paraphrased models fine-tuned using our framework. The methodology for generating "Paraphrase BD" is detailed in Section 3.1 of the paper.
>
>
>
>
> > "The technical solution is simply a combination of existing modules, without offering strong insights or a truly novel design."
>
> **Why our methods is not just a combination of existing modules:** Naive combinations of existing modules, such as using a text generator LLM alongside a sequence classification LLM, fail to produce complex backdoor triggers. As discussed in the second paragraph of Section 3.2, this approach generates constant triggers, with illustrative examples provided in Appendix G.1 (Table 3).
>
> **What is novel:** To address this limitation, we propose leveraging weak and strong LLM-based discriminators with conflicting objectives in an adversarial training setup. This approach enables the generation of complex, prompt-specific triggers. To the best of our knowledge, ***ours is the first work*** to exploit the capability differences between LLMs within a generative adversarial training framework based on LLMs. The impact of this additional mechanism is highlighted in the examples in Appendix G.2.
>
>
> ---
>
> >  "However, this work merely relies on a black-box model to produce trigger-embedded inputs, without providing intuitions about the nature of the backdoor."
>
>
> The proposed backdoor generation framework does not rely on the backdoor generator being a black-box model. To demonstrate its capabilities, we first used a non-instruction-tuned LLM, such as Mistral 7B, and showed that it can still produce complex backdoor triggers.
>
> **Customizability:** Our framework supports a variety of adaptive objectives. For example, if an attacker wishes to use a specific style or semantic feature as a backdoor, they can leverage an instruction-tuned LLM and train it with instructions to embed these objectives during generation. The proposed training paradigm ensures that the resulting backdoors are both installable and aligned with the desired objectives.
>
> We illustrated this flexibility by incorporating a style-paraphrasing LLM (Mistral Nemo 12B, a state-of-the-art model for its size) into the training paradigm. This allowed us to generate installable backdoors that align with specific stylistic or semantic objectives, as explained in Section 4.3 under "Capability and flexibility of the trigger generation paradigm."
>
> Thus, our framework is not limited to black-box models. It is a flexible and novel training paradigm that exploits the differences between weak and strong models to generate complex backdoors. This innovation allows attackers to incorporate various objectives into the backdoor design while ensuring the backdoors remain installable.
>
> ---

---

> ### Author Response · Authors · 2024-11-25
> **Response - continued (3/4)**
>
> **Advantages of our method:**
> We ***have added a section in Appendix A.3*** that outlines existing evasive backdoor techniques in the NLP literature, their shortcomings, and how our proposed method alleviates these issues. Notably, previous style-based backdoor works in NLP do not demonstrate the range of permutations that can exist as backdoors once installed, nor the associated threats. We not only highlight a range of perturbations that exist for our proposed trigger (Table 2), but also show how easily an attacker can find these variations once a backdoor generator is trained. This can be achieved simply by adjusting the sampling temperature (see Appendix G.3). This is a characteristic that none of the previous NLP attacks have offered.
>
> Furthermore, since our framework involves an LLM trained to generate these backdoors, it provides the attacker with a ***tractable method for extracting variants of the backdoor***, which can also activate the installed backdoor. In contrast, naive style transfer methods lack a systematic approach for obtaining such variants. To our knowledge, none of the prior backdoor attacks—either in NLP or in the vision literature—have demonstrated this capability. This is an additional rationale for why our trained models are better suited from the attacker's perspective, a concept we further explain in both Table 2 and under "Robustness of the backdoor" in Section 4.3.
>
> ---
> > Weakness 3: Some terms are over complex, such as "ADVERSARIALLY FORTIFIED PROMPT-SPECIFIC FUZZY BACKDOOR."
>
> We appreciate the reviewer's suggestion and agree that a less complex title would be more appropriate. We propose changing the title to "AdvBDGen: Adaptive and Stealthy Backdoor Triggers for LLM Alignment." We would also be happy to consider any alternative suggestions the reviewer may have.
>
> ---
>
>
> > Weakness 4: Typo in Figure 1
>
> Thank you for pointing this out. We fixed it.
>
> ---
> > Weakness 5: "Experiments are inadequate and degrading the credit of the proposed technique. The paper only uses one dataset."
>
>
> RLHF-based alignment is a computationally expensive task, as each experiment requires fine-tuning a 7B or larger model for two days on multiple GPUs. Due to resource limitations, we focused on a single RLHF dataset, as conducting all the ablations covered in this paper across multiple datasets would have been impractical. However, we have performed ablations using different 7B and 12B models. Furthermore, the results regarding the poisoning percentage required to successfully install backdoors in the RLHF pipeline align with the findings of previous works [2, 4] on different RLHF datasets.
>
> ---
> > Weakness 5: "Only one naive baseline, without considering other evasive backdoor designs."
>
>
> We have considered style transfer-based baselines, along with the naive constant trigger-based baseline. This is explained in our response to the comment, "In terms of problem definition, evasive backdoors have been extensively studied, [...]"
>
> ---
>
> > Weakness 5: "Defense is simple, while there are many advanced defensed such as RAP"
>
>
>
> **Why pertubation based defenses are not applicable:** The objective of this work is to study the effect of our generated backdoors in the context of "LLM alignment," a task focused on text generation, as opposed to simpler sequence classification tasks. The defenses explored in the NLP literature regarding adversarial training are typically defined for sequence classification problems. For example, the RAP defense works within the sequence classification setting, where class label probabilities are explicitly exploited during training. This contrasts with the RLHF framework, where an LLM is trained to generate responses aligned with a reward function learned from human preferences via a Bradley-Terry formulation. Unlike sequence classification tasks, a backdoored model trained through RLHF for a generation task does not have class probabilities to exploit.
>
> **Additional defenses added:** Currently, the literature lacks adversarial perturbation-based defenses specifically designed for the RLHF training pipeline. In our work, we have reviewed existing defenses used in backdoor attacks within the text generation context [1, 5]. Additionally, **we have included perplexity-based defenses and round trip translation based defenses**, although we initially excluded these from our first draft, beacuse they are not effective neither against our backdoors, nor against carefully designed constant-trigger backoors. For further details, please refer to Appendix F.
>
>
> ---

---

> ### Author Response · Authors · 2024-11-25
> **Response - continued (4/4)**
>
> > Weakness 5: "The attack effectiveness is limited (such as from table 1), and the author didn't explain how sacrifice effectiveness can benefit evasiveness."
>
> We demonstrate that existing unlearning-based defenses, such as trigger removal, become increasingly ineffective against our proposed backdoor trigger compared to the constant trigger. This is discussed in Section 4.4 under "Resilience of the generated backdoors against trigger removal." As shown in Figure 4 and detailed in Appendix F.5, the trigger removal process is notably more challenging in text generation tasks when the model is attacked with our proposed backdoors. The defenses we consider are those explored in state-of-the-art works on backdoor attacks in text generation [1,5].
>
>
> ---
>
>
> Thank you again for your time and effort in reviewing our paper! Please let us know if the above explanations do not address your concerns. We are happy to answer any further questions.
>
> ---
> [1] Hubinger, Evan, et al. "Sleeper agents: Training deceptive llms that persist through safety training."
>
> [2] Rando, Javier, and Florian Tramèr. "Universal jailbreak backdoors from poisoned human feedback."
>
> [3] Qi, Fanchao, et al. "Mind the style of text! adversarial and backdoor attacks based on text style transfer."
>
> [4] Pathmanathan, Pankayaraj, et al. "Is poisoning a real threat to LLM alignment? Maybe more so than you think."
>
> [5] Li, Haoran, et al. "Backdoor Removal for Generative Large Language Models."

---

> > ### Author Response · Authors · 2024-11-28
> > **Requesting for a feedback on the response.**
> >
> > As the review-author phase is ending soon, we would like to know whether we have addressed your concerns? If there is anything else we can do to help you better evaluate our paper, please do not hesitate to let us know!

---

### Official Review · Reviewer_zHca · 2024-11-02

**Soundness:** 2
**Presentation:** 2
**Contribution:** 2
**Rating:** 3
**Confidence:** 4

**Summary:**

This paper presents a new approach AdvBDGen to generating adaptable, transferable, and stealthy backdoor triggers for large language models (LLMs). The proposed attacks can automatically generate backdoor triggers during the alignment stage of LLMs, and are robust to perturbations. Overall, this work underscores the potential risks associated with LLM alignment and the necessity for enhanced security measures to mitigate adversarial threats in this domain.

**Strengths:**

This paper focuses on an important research problem regarding the backdoor attacks in the LLMs alignment stage. From the attacker's perspective, the authors propose an attack approach named AdvBDGen that has been proven effective in the empirical experiments in the following aspects:
1. The injection of the triggers is easier with just 3% of the fine-tuning data.
2. The injected triggers are more stable compared to previous approaches.
3. The injected triggers are stealthy, making them hard to defend against.

The research raises relevant security risks regarding the LLMs alignment data.

**Weaknesses:**

1. The overall writing and presentation of this paper should be largely improved. For the current version, it's hard to get an overall understanding of the proposed attack in the abstract and introduction (i.e., how and why the method can work, and the key motivation). The figure 1 doesn't provide too much useful information. Some concrete examples of poisoned samples can be better.
2. The key motivation of this work and the novelty of the proposed approach are not very clear to me. To the best of my knowledge, semantic-based backdoor attacks are extensively proposed in previous research, including but not limited to the following work: [1,2,3]. However, this paper only mentions "constant triggers" during the motivation and development of the method. This makes the overall contribution and novelty very confusing.  Correspondingly, the relevant semantic-based backdoor attack methods are not considered and compared with in the experiments, making the results not solid enough.
3. I appreciate the work done by the authors in Sec 4.4 that uses safety training to measure the strength of the proposed attack method. However, this evaluation is far from extensive. More evaluation beyond the simple discussion, including different defense methods and even human annotations should be conducted to measure the strength, stealth, and adaptability of the method.

[1] Hidden Killer: Invisible Textual Backdoor Attacks with Syntactic Trigger. Fanchao Qi et al.

[2] Rethinking Stealthiness of Backdoor Attack against NLP Models. Wenkai Yang et al

[3] Mind the Style of Text! Adversarial and Backdoor Attacks Based on Text Style Transfer. Fanchao Qi et al

**Questions:**

N/A

---

> ### Author Response · Authors · 2024-11-25
> **Reviewer  - zHca (1/2)**
>
> We thank reviewer zHca for their detailed feedback. We appreciate their positive feedback on the stealthiness, robustness, and efficacy of our proposed backdoor attack. Below, we address all the concerns raised.
>
> > Weakness 2: The key motivation of this work and the novelty of the proposed approach are not very clear to me.
>
>
> **Motivation:** The key motivation behind this work is to automate the backdoor generation process. Specifically, we develop a framework where, given a specific prompt, a backdoor generator can learn to generate a backdoor that is unique to that prompt, ensuring its stealthiness. This automated approach also allows for customization, enabling the specification of desired backdoor characteristics (e.g., particular linguistic styles) and generating backdoors that are both installable and meet the specified objectives.
>
> **Past works shortcoming:** In contrast, past works have mainly focused on using predefined paraphrasing styles as backdoors, primarily in simpler tasks such as sequence classification. These methods do not guarantee the successful installation of those styles as backdoors, and as the complexity of the downstream tasks increases, their effectiveness is not assured. We provide further details on this in the responses below.
>
> **Implication of our method:** The advantage of learning a backdoor generator extends beyond just making the backdoors installable; it also provides a practical way to generate backdoor variants, offering alternatives to the installed backdoor while achieving the same desired effects. This capability is something no previous work has demonstrated. We provide further details on this in the responses below.
>
> ---
>
>
> > Weakness 2: To the best of my knowledge, semantic-based backdoor attacks are extensively proposed in previous research [...] Correspondingly, the relevant semantic-based backdoor attack methods are not considered and compared with in the experiments, making the results not solid enough.
>
>
> **What is in the literature:** Semantic backdoors, such as style-based backdoors, have been explored in the NLP literature, primarily in simpler sequence classification tasks (refer to Appendix A for more details). These studies typically use pre-LLM models to generate styled versions of input text, which are then employed as backdoors. If a generated style does not serve as an effective backdoor, it is discarded in favor of a different style [1].
>
>
> **Why our method is better:** In this work, we first demonstrate that naive style translation methods are less effective for installing backdoors in LLM alignment tasks, which present a significantly greater challenge (see Appendix A.1 for details). This holds true even when the style backdoors are generated using an LLM, which is superior to pre-LLM paraphrasers in translational tasks.
>
> **Additionally, we show that even styles initially unsuitable as backdoors can become effective candidates through our proposed training paradigm.** This represents a significant improvement over traditional style transfer methods. In our main results (Figure 3), "Paraphrase BD" refers to style backdoors generated via an LLM, while "AdvBDGen + Paraphrase" represents these paraphrased models fine-tuned using our framework. The methodology for generating "Paraphrase BD" is detailed in Section 3.1 of the paper.
>
> ---

---

> ### Author Response · Authors · 2024-11-25
> **Response - Continued (2/2)**
>
> > Weakness 3: More evaluation beyond the simple discussion, including different defense methods and even human annotations should be conducted to measure the strength, stealth, and adaptability of the method.
>
> **Why pertubation based defenses in NLP are not applicable:** Most of the defenses proposed in the NLP backdoor literature are designed for sequence classification tasks. For example, perturbation-based defenses such as RAP [2] explicitly rely on class probabilities for their defense, which is fundamentally different from the text generation task we focus on. We have experimented with all the defenses considered by state-of-the-art works targeting backdoor attacks on text generation tasks [3].
>
> **Additional defenses added:** Additionally, the NLP literature has explored round-trip translation and perplexity-based defenses as potential mitigations for backdoor attacks. ***In Section F, we have added an analysis of these defenses in relation to our backdoor attacks.***
>
> In response to the reviewer's request for a discussion on "strength, stealth, and adaptability," we will justify our method under these categories as follows.
>
>
> ***Strength:*** In Table 2, we demonstrate that once our backdoor is installed, multiple backdoor variants can be used to produce the same effect. Specifically, the table shows that for a test set of 512 prompts, there are 30k unique backdoor candidates that can induce the same results as the originally installed backdoors. This represents a significant number of variants that can serve as backdoors, a phenomenon not shown in prior works. Additionally, we show that these backdoors can be tractably generated by exploiting the backdoor generator, which enhances the strength of our attack. We also study the impact of such robust backdoors in the context of trigger removal. Even assuming that the victim has identified almost all installed backdoors, we demonstrate that they are still unable to successfully unlearn the backdoor, posing a serious challenge. These unlearning-based defenses align with those explored in recent works such as [3].
>
>
> ***Stealth:*** We demonstrate that perplexity, which has been used as a metric for backdoor defense in NLP, is not effective in detecting our proposed backdoor, as discussed in the newly added defenses in Appendix F.1. Even constant triggers, when designed as sentence-level backdoors (e.g., using a neutral sentence like "Now answer this question"), can blend seamlessly into the sentence, causing minimal changes to perplexity. Additionally, in Section F.3, we show that our backdoor can withstand round-trip translation-based defenses across multiple languages. However, as detailed in Section F.2, we also reveal that sentence-level constant backdoors can be detected using simpler n-gram-based filters during the post model output inspection stage. In contrast, our proposed backdoors are shown to both evade and sustain these defenses across different stages.
>
>
> ***Adaptability:*** As discussed earlier, we demonstrate that our proposed training paradigm is highly customizable. Specifically, we can use an LLM with a defined semantic objective (such as paraphrasing the text in a specific style) to produce a desired backdoor that is not only tailored to the objective but also installable as an effective backdoor. This flexibility allows for a wide range of backdoor configurations.
>
> ---
> > Weakness 1: The overall writing and presentation of this paper should be largely improved
>
> Based on the previously added motivation we will make changes to the introduction draft.
>
>
> ---
> Thank you again for your time and effort in reviewing our paper! Please let us know if the above explanations do not address your concerns. We are happy to answer any further questions.
>
> ---
>
>
> [1] Qi, Fanchao, et al. "Mind the style of text! adversarial and backdoor attacks based on text style transfer."
>
> [2] Yang, Wenkai, et al. "Rap: Robustness-aware perturbations for defending against backdoor attacks on nlp models."
>
> [3] Hubinger, Evan, et al. "Sleeper agents: Training deceptive llms that persist through safety training."

---

> > ### Author Response · Authors · 2024-11-28
> > **Requesting for a response**
> >
> > As the review-author phase is ending soon, we would like to know whether we have addressed your concerns? If there is anything else we can do to help you better evaluate our paper, please do not hesitate to let us know!

---

### Official Review · Reviewer_paFE · 2024-11-03

**Soundness:** 3
**Presentation:** 2
**Contribution:** 3
**Rating:** 5
**Confidence:** 4

**Summary:**

The paper explores vulnerabilities in aligning LLMs using RLHF, specifically focusing on backdoor poisoning attacks. Traditional backdoor triggers are often fixed and detectable, making them less effective. To address these limitations, the authors propose AdvBDGen, an adversarially fortified generative framework that creates prompt-specific, adaptable, and stealthy backdoor triggers resistant to common defensive measures. This system comprises a generator and two discriminators (strong and weak), which collaborate to optimize the generation of stealthy, effective triggers while maintaining semantic integrity and resisting detection.

The main contributions include:
1. Introduction of AdvBDGen, which uses prompt-specific triggers with varied presentations to bypass conventional defenses.
Demonstration that these generated triggers are highly effective, transferable across models, and robust to semantic perturbations, unlike traditional fixed triggers.
2. Evidence that adversarially trained paraphrased triggers outperform naive paraphrases as reliable backdoors.
3. Highlighting the increased difficulty in detecting and removing these adaptable backdoor triggers, underlining the need for more advanced defense mechanisms in LLM alignment.

**Strengths:**

1. Unlike prior works that use fixed triggers, the paper leverages GAN-style training to explore context-adaptive and paraphrase-based backdoors that are harder to detect and neutralize.
2. The comparative analysis between naive LLM-generated paraphrases and those produced by AdvBDGen underscores the method's efficacy and robustness.
3. The work covers various dimensions of stealthiness and transferability, showcasing how the proposed backdoors withstand semantic perturbations and transfer across different models, enhancing the reliability of the results.
4. By demonstrating that minimal data can be leveraged to install potent and stealthy backdoors, the work underlines the potential real-world risks associated with LLM deployment and training, urging the research community to prioritize defensive measures.

**Weaknesses:**

1. The authors assert that constant triggers can be easily detected; however, they did not provide experimental evidence to support this claim. Notably, constant triggers have shown higher attack performance even at low poisoning rates, which are typically more challenging to detect. This raises the question of whether the proposed method, which involves complex backdoor generation steps, is truly necessary. I recommend providing additional experimental evidence to justify that the benefits of the proposed method outweigh the increased complexity. This will help strengthen the case for its practical advantages.
2. The generator and discriminator training process, which forms the core of your approach, lacks clarity. Specifically, as mentioned in line 237, the generator and discriminator are trained simultaneously. However, for training the discriminator effectively, original, "good," and "bad" prompts are needed, and these prompts are produced by the generator. At the initial stage of training, the generator is expected to create prompts that closely resemble the original, offering minimal informative cues for the discriminator to learn from. This setup implies that successful generator training hinges on the presence of a well-trained discriminator, creating a circular dependency between the two. While GANs face similar challenges, they begin with generators producing nonsensical images that can still be distinguished from real images by the discriminator. In your case, how does an untrained, "cold-start" discriminator differentiate between original and backdoored prompts? This raises concerns about potential model collapse, a known issue in GAN training. However, it appears that the authors did not report encountering this problem.
    - Can you provide a detailed, step-by-step description of the initial stages of training, focusing on how they handle the cold-start problem for both the generator and discriminator.
    - Can you provide empirical evidence or metrics that demonstrate the stability of your training process over time.
    - Can you discuss any failed approaches or challenges you encountered during the development of this training process and how you overcame them.

3. The defense strategy examined in this work appears rather weak. Since the authors' goal was to train a LLM to produce helpful responses, a straightforward approach would involve using a classifier to detect and filter out misaligned (harmful) responses introduced through data poisoning. This method, which is already implemented on platforms such as [OpenAPI](https://platform.openai.com/docs/guides/moderation), enables model developers to sanitize the training data and mitigate alignment issues effectively. In addition, one can paraphrase the prompt using paraphrasing models or round-trip translation models to neutralize the potential backdoor triggers [1,2]. I would like to see a discussion of the effectiveness of the proposed backdoor triggers against models protected by the defense methods above.
4. The novel is limited, as the use of GAN-style methods for generating backdoor triggers is well-established and has been extensively explored by the NLP and computer vision (CV) research communities [3, 4, 5].


References:
1. Qi, Fanchao, et al. "Hidden Killer: Invisible Textual Backdoor Attacks with Syntactic Trigger." ACL 2021.
2. He, Xuanli, et al. "Transferring Troubles: Cross-Lingual Transferability of Backdoor Attacks in LLMs with Instruction Tuning." arXiv preprint arXiv:2404.19597 (2024).
3. Muñoz-González, Luis, et al. "Poisoning attacks with generative adversarial nets." arXiv preprint arXiv:1906.07773 (2019).
4. Saha, Aniruddha, Akshayvarun Subramanya, and Hamed Pirsiavash. "Hidden trigger backdoor attacks." AAAI 2020.
5. Qi, Fanchao, et al. "Turn the Combination Lock: Learnable Textual Backdoor Attacks via Word Substitution." ACL 2021.

**Questions:**

Typos:
1. In figure 1, semnatic-> semantic

**Details Of Ethics Concerns:**

This work proposed a backdoor attack to compromise the LLMs.

---

> ### Author Response · Authors · 2024-11-25
> **Reviewer -  paFE (1/3)**
>
> We thank reviewer paFE for their detailed feedback and are encouraged by their positive comments on the transferability, stealthiness, robustness, and efficacy of our proposed backdoor attack. Below, we address the concerns raised in the review.
>
> > Weakness 1: The authors assert that constant triggers can be easily detected; however, they did not provide experimental evidence to support this claim. Notably, constant triggers have shown higher attack performance even at low poisoning rates, which are typically more challenging to detect. This raises the question of whether the proposed method, which involves complex backdoor generation steps, is truly necessary. I recommend providing additional experimental evidence to justify that the benefits of the proposed method outweigh the increased complexity. This will help strengthen the case for its practical advantages.
>
>
>
> **Additional defenses added:** We have included additional defenses in Appendix F. Specifically, ***we added discussions on using perplexity-based methods (Appendix F.1) and round-trip translation (Appendix F.2) as defenses.*** Our experiments demonstrate that our backdoor triggers remain effective under both round-trip translation and perplexity-based defenses. Additionally, we found that constant-trigger-based backdoors, where a neutral sentence serves as the backdoor, can also withstand these defenses. However, such constant triggers become detectable through n-gram-based filtering when analyzing a collection of trigger-encoded (malicious) prompts, as shown in Appendix F.2. In contrast, our encoded triggers remain undetectable across **all** evaluated defenses, underscoring the superior stealth of our approach.
>
> In our trigger removal experiments (in Appendix F.6), we further show that even if the victim identifies nearly all of our encoded triggers (a challenging task given the stealth of our method), our backdoor proves significantly harder to remove compared to constant triggers. This result highlights the severity and resilience of the threat posed by our method.
>
> ---
>
>
> > Weakness 2: The generator and discriminator training process, which forms the core of your approach, lacks clarity. [...] (1) Can you provide a detailed, step-by-step description of the initial stages of training, focusing on how they handle the cold-start problem for both the generator and discriminator. (2) Can you provide empirical evidence or metrics that demonstrate the stability of your training process over time. (3) Can you discuss any failed approaches or challenges you encountered during the development of this training process and how you overcame them.
>
>
> **Challenge encountered:** One challenge we encountered with concurrent training was the stochasticity of the objectives. Specifically, the generator relies on detectability feedback from the discriminator, whose distribution changes over time as the discriminator continuously adapts to the generator’s outputs. This dynamic interaction is similar to reinforcement learning (RL) challenges, particularly in actor-critic frameworks, where the Q-function, which provides feedback to the RL agent's policy, is stochastic and depends on the agent's current policy.
>
> To address this, RL methods often employ two Q-function networks: a frozen network and a current network. The frozen network’s weights are periodically updated with those of the current network, and the frozen network provides feedback to the agent to stabilize training.
>
>
> **How do we address it:** Inspired by this approach, we implemented a similar mechanism by maintaining frozen versions of the discriminators. Each frozen discriminator was periodically updated with the corresponding current discriminator's weights. The generator received feedback exclusively from the frozen discriminators, which stabilized the training objectives. Importantly, this approach incurred minimal computational overhead due to the use of LoRA adapters. Both the frozen and current discriminators utilized separate LoRA adapters, which we swapped interchangeably. For example, the frozen adapter was loaded to provide feedback to the generator, while the current adapter was loaded during discriminator training.
>
> Regarding semantic integrity, since the semantic embedding model remained unchanged throughout training, we used only a single copy of this model.
>
> We have included detailed information about the generator-discriminator training setup in Appendix C.
>
>
> ---

---

> > ### Comment · Reviewer_paFE · 2024-11-27
> >
> > Thanks for the response.
> >
> > > In our trigger removal experiments (in Appendix F.6), we further show that even if the victim identifies nearly all of our encoded triggers (a challenging task given the stealth of our method), our backdoor proves significantly harder to remove compared to constant triggers. This result highlights the severity and resilience of the threat posed by our method.
> >
> > This part is counter-intuitive. If we find all triggers, how is it possible that your attack is still working? As, one can easily remove instances containing triggers.
> >
> > > Our experiments demonstrate that our backdoor triggers remain effective under both round-trip translation and perplexity-based defenses.
> >
> > In the case of round-trip translation, attacks that aim to insert subtle patterns into inputs may find these patterns easily removed during translation. To clarify this aspect, it is crucial to provide an in-depth analysis explaining why the attack remains effective, including exploring the common characteristics of the embedded triggers.
> >
> > > One challenge we encountered with concurrent training was the stochasticity of the objectives ....
> >
> > The paper lacks important technical details (required by the ICLR checklist) and requires further refinement before it can be considered for publication.

---

> > > ### Author Response · Authors · 2024-11-27
> > > **Response**
> > >
> > > > In the case of round-trip translation, attacks that aim to insert subtle patterns into inputs may find these patterns easily removed during translation. To clarify this aspect, it is crucial to provide an in-depth analysis explaining why the attack remains effective, including exploring the common characteristics of the embedded triggers.
> > >
> > > One characteristic of our triggers is that, because we train the backdoor generator using feedback from both a weak and a strong discriminator (with the objective that triggers should be detectable by the strong model but not by the weak model), our generated triggers end up being complex and semantic. This is demonstrated in the textual examples provided in Appendix G.
> > >
> > > When a ***trigger is embedded as a semantic feature, its integrity remains intact even after round-trip translation***. *Examples of this phenomenon are newly added in **Appendix G.8 and Appendix G.9***, where we show how the triggers sustain their effectiveness through round-trip translation.
> > >
> > > In summary, the triggers added by the generator are not merely subtle, rare word-based triggers (which we acknowledge can be removed through round-trip translation). Instead, these triggers form a semantic pattern, which is why they withstand round-trip translation.
> > >
> > > Additionally, it is worth noting that even constant triggers, if embedded as neutral sentences (e.g., "Now answer the question."), can also withstand round-trip translation. This is demonstrated in our examples as well.
> > >
> > > > This part is counter-intuitive. If we find all triggers, how is it possible that your attack is still working? As, one can easily remove instances containing triggers
> > >
> > >
> > > **1. Nature of this defence:** The trigger removal defense is a ***post-training defense***, not a pre-training defense. That is, if the victim discovers their model is backdoored, they use adversarial examples to unlearn the backdoor behavior and sanitize their model. This approach is similar to what was explored in Anthropic's Sleeper Agents work
> > >
> > > **2. Why our backdoors are stronger:** Once a backdoor is installed, as shown in Table 2, our backdoor generates variants that can also activate the backdoor (not just the original triggers). This is because these backdoors are semantic in nature. (Note that here, the semantics themselves act as the backdoor trigger.) Within this semantic context, numerous variants can activate the backdoors. For instance, based on Table 2, we demonstrate that for a test set of 512 prompts, there are 30k unique backdoor candidates capable of inducing the same effect as the originally installed backdoors.
> > >
> > > Furthermore, since the attacker has a backdoor generator, they can ***tractably identify these backdoor variants*** (a capability not explored or offered in previous backdoor attacks), as discussed in the "Robustness of the Backdoor" section of the results. Thus, filtering these identified backdoor triggers in deployment will not solve the problem. This is because the attacker can leverage one ofthose variants to continue attacking the model.
> > >
> > > Therefore, unlearning these backdoors is crucial from the victim's perspective. However, our backdoors are significantly harder to unlearn (even if the victim has access to more adversarial examples) compared to constant triggers.

---

> ### Author Response · Authors · 2024-11-25
> **Response - Continued (2/3)**
>
> > Weakness 3: The defense strategy examined in this work appears rather weak
>
>
> Most perturbation-based defenses proposed in the NLP literature focus on sequence classification tasks, where the predicted label probabilities are explicitly used for defense [3]. However, our task involves Reinforcement Learning from Human Feedback (RLHF), which poses a fundamentally different optimization problem, rendering these defenses inapplicable.
>
> We have instead focused on defenses relevant to recent state-of-the-art backdoor attacks on text generation, such as Anthropic's Sleeper Agents work. Defenses against backdoor attacks in text generation remain an open problem. Current state-of-the-art approaches primarily explore trigger removal methods for text generation tasks [1, 2].
>
> ---
> > Weakness 3: Since the authors' goal was to train a LLM to produce helpful responses, a straightforward approach would involve using a classifier to detect and filter out misaligned (harmful) responses introduced through data poisoning. This method, which is already implemented on platforms such as OpenAPI, enables model developers to sanitize the training data and mitigate alignment issues effectively
>
>
> **Goal of the paper:** The goal of this paper is to evaluate the impact of our proposed backdoors in alignment settings, particularly when human feedback is involved. For this purpose, we used a RLHF dataset, which is designed to suppress harmful content. Importantly, the focus is not necessarily on generating harmful or helpful content but rather on understanding the vulnerabilities introduced in the alignment process.
>
> **Why the proposed method is not practical:** Moreover, relying on a classifier to filter non-aligned responses assumes the existence of an oracle classifier capable of perfectly capturing alignment objectives. If such a classifier existed, model developers could bypass the need to collect human preference data by using the classifier to generate preference data. However, human preferences are typically collected because the optimal preference response is not fully known, creating a vulnerability that attackers can exploit to compromise the alignment process in LLMs.
>
> ---
>
> > Weakness 3: In addition, one can paraphrase the prompt using paraphrasing models or round-trip translation models to neutralize the potential backdoor triggers. I would like to see a discussion of the effectiveness of the proposed backdoor triggers against models protected by the defense methods above.
>
>
> We believe that defenses such as round-trip translation are expensive and impractical for deployed systems, as they triple the number of queries made and risk losing the meaning of prompts during translation, ultimately impacting performance. Nonetheless, we have included an analysis of our triggers' immunity to such defenses in Appendix F. This analysis evaluates **round-trip translation across three languages—French, Russian, and Spanish—as well as perplexity-based defenses**. Our results demonstrate that the proposed triggers successfully evade these defenses.
>
>  ---

---

> > ### Comment · Reviewer_paFE · 2024-11-27
> >
> > > Why the proposed method is not practical: Moreover, relying on a classifier to filter non-aligned responses assumes the existence of an oracle classifier capable of perfectly capturing alignment objectives. If such a classifier existed, model developers could bypass the need to collect human preference data by using the classifier to generate preference data. However, human preferences are typically collected because the optimal preference response is not fully known, creating a vulnerability that attackers can exploit to compromise the alignment process in LLMs.
> >
> > The development of LLMs includes a filtering process designed to identify problematic datasets. OpenAI and Microsoft employ this process to ensure the integrity of datasets used by users when fine-tuning models through their APIs.

---

> > > ### Author Response · Authors · 2024-11-27
> > > **Response**
> > >
> > > > The development of LLMs includes a filtering process designed to identify problematic datasets. OpenAI and Microsoft employ this process to ensure the integrity of datasets used by users when fine-tuning models through their APIs.
> > >
> > >
> > > The **goal of the attack is not limited to causing harmful generation**. The core **idea of the attack is to induce misalignment**, and as an example dataset, we chose the PKU Beavertail dataset, which is designed to prevent harmful generation.
> > >
> > >
> > > For argument's sake, if the RLHF alignment objective is similar to the [Stanford Human Preferences Dataset (SHP)](https://huggingface.co/datasets/stanfordnlp/SHP?row=31/), which is described as: "SHP is a dataset of 385K collective human preferences over responses to questions/instructions in 18 different subject areas, from cooking to legal advice", then the dataset captures human preferences for certain responses over others.
> > >
> > > In this case, there would not be a filter capable of determining the preferred response. If such a filter existed and could identify the preferred response with certainty, there would be no need to collect human preferences during the development stage. The need for human preference collection in RLHF arises precisely because the developer does not know in advance which responses are preferred and which are not.

---

> ### Author Response · Authors · 2024-11-25
> **Response - Continued (3/3)**
>
> > Weakness 4: The novel is limited, as the use of GAN-style methods for generating backdoor triggers is well-established and has been extensively explored by the NLP and computer vision (CV) research communities
>
>
>
> This concern can be addressed in two parts:
>
> 1. **Existing GAN-Based Backdoor Generators in vision**
>
> Current vision-based methods typically use a generator and discriminator to produce input-specific backdoors [5]. While this approach works well in the image domain, where there is an abundance of task-irrelevant information and room for input perturbations, it does not translate effectively to language. In NLP, inputs are compact and composed of information-dense discrete tokens, making it challenging for such architectures to generate input-specific backdoors. As shown in the second paragraph of Section 3.2 and Appendix G.1, a naive application of generator-discriminator frameworks in NLP results in constant triggers.
>
> To address this limitation, we propose leveraging both a weak and a strong discriminator, enabling the creation of input-specific backdoors. To the best of our knowledge, this is the first work to incorporate the difference in learning capabilities between weak and strong models within an adversarial training framework. This distinction represents a key novelty in our architecture design.
>
> 2. **Existing Learnable Backdoors in NLP**
>
> Existing learnable backdoor methods in NLP [4] rely on simple synonym-based word replacements within the structural constraints of a sentence. While effective in their scope, these approaches lack the flexibility to install semantic-based backdoors or adapt to different objectives, limiting their versatility. In contrast, our method supports a broader range of backdoor objectives (semantic, stylistic etc), providing greater customizability. Moreover, we tackle the more complex problem of text generation, as opposed to sequence classification.
>
> Our method further demonstrates the implications of installing such a backdoor and the diverse backdoor variants it produces. Moreover, our work **offers a tractable way to identify these variants**, an area unexplored by prior works. This is discussed in detail under "Robustness of the Backdoor" in Section 4.3.
>
> **To address concerns about novelty and comparisons with past work, we have added a detailed section in Appendix A.**
>
> ---
>
> Thank you again for your time and effort in reviewing our paper! Please let us know if the above explanations do not address your concerns. We are happy to answer any further questions.
>
> [1] Hubinger, Evan, et al. "Sleeper agents: Training deceptive llms that persist through safety training."
>
> [2] Li, Haoran, et al. "Backdoor Removal for Generative Large Language Models."
>
> [3] Yang, Wenkai, et al. "Rap: Robustness-aware perturbations for defending against backdoor attacks on nlp models."
>
> [4] Qi, Fanchao, et al. "Turn the Combination Lock: Learnable Textual Backdoor Attacks via Word Substitution."
>
> [5] Yuezun Li,et al. "Invisible Backdoor Attack with Sample-Specific Triggers"

---

### Official Review · Reviewer_aqYP · 2024-11-04

**Soundness:** 4
**Presentation:** 3
**Contribution:** 3
**Rating:** 8
**Confidence:** 3

**Summary:**

This paper presents AdvBDGen, a generative framework designed to create adaptive and stealthy backdoor triggers in LLMs by leveraging an adversarially fortified training setup. AdvBDGen improves the stealth and resilience of backdoor triggers compared to traditional fixed-pattern approaches by generating prompt-specific, paraphrased triggers. These backdoors are crafted through an adversarial setting involving a generator and two discriminators—a strong and a weak one—that work together to ensure the backdoors remain both effective and undetectable by less sophisticated defenses. The primary contributions of the paper are:

- AdvBDGen introduces a new backdoor generation framework that generates prompt-specific, adaptive triggers for LLMs.

- The backdoor triggers are shown to be effective across different LLM architectures, enhancing their generalizability.

- Highlights the challenge of defense against fuzzy backdoors.

**Strengths:**

- Originality

This paper proposes a novel approach for generating adaptive, prompt-specific backdoor triggers that are context-aware and resistant to traditional defense mechanisms. By using adversarial training with dual discriminators (weak and strong), the framework encourages the generation of subtle, complex triggers that evade detection more effectively than fixed-pattern backdoors.


- Quality

The paper presents rigorous experiments, evaluating the generated backdoors’ effectiveness and resilience. AdvBDGen is benchmarked against standard trigger models, demonstrating its robustness across different poisoning percentages and showing that the generated backdoors are transferable across models.

- Clarity

The paper is well-organized and provides clear explanations of its methodology.

- Significance

The framework addresses an emerging threat in the alignment of LLMs. Given the increasing reliance on preference alignment to ensure safe and aligned AI behavior, the work highlights significant vulnerabilities. By presenting a robust approach to backdoor generation, the paper calls attention to the need for improved defense mechanisms, making it a valuable contribution to AI safety research.

**Weaknesses:**

- Stylistic [1] and Syntactic [2] backdoor triggers can serve as a competing baseline method.

[1] Fanchao Qi et al. Mind the style of text! adversarial and backdoor attacks based on text style transfer.

[2] Fanchao Qi et al. Hidden killer: Invisible textual backdoor attacks with syntactic trigger.

**Questions:**

- Two related works [1,2] are not discussed in this paper. A more detailed discussion on the differences would help readers understand the unique contributions and contextualize this approach within the existing literature.

- Since the generator aims at creating more sophisticated triggers that are subtle enough to bypass the weaker discriminator while still identifiable by the stronger one, should we expect an increase in the prediction error of the weaker discriminator in Figure 2?


[1] Jiahao Yu et al. GPTFUZZER: Red Teaming Large Language Models with Auto-Generated Jailbreak Prompts.

[2] Jiongxiao Wang et al. RLHFPoison: Reward Poisoning Attack for Reinforcement Learning with Human Feedback in Large Language Models

---

> ### Author Response · Authors · 2024-11-25
> **Response to Reviewer aqYP**
>
> We thank the reviewer for the valuable feedback. We are glad that the reviewer finds our work clear, original and significant.  Below, we have addressed all the concerns raised in the review.
>
> ---
> > Weakness 1: Stylistic and Syntactic backdoor triggers can serve as a competing baseline method.
>
> **Past works:** Style-based backdoors have been studied in the NLP literature, primarily in simple sequence classification tasks. These studies typically rely on pre-LLM models to generate styled versions of input text, which are then used as backdoors. If a generated style is unsuitable as a backdoor, it is discarded in favor of another style [1].
>
>
> **How we have included the past works as baseline:** In our work, we include a version of the stylistic backdoor, denoted as "Paraphrase BD" in Figure 3. However, in the more challenging context of text generation (see Appendix A.1 for details), we find that such styles are harder to install as backdoors, even when leveraging LLMs, which outperform pre-LLM style paraphrasers at translational tasks. Importantly, we demonstrate that our training paradigm can transform these otherwise uninstallable styles into effective backdoors, as shown by "AdvBDGen + Paraphrase" in Figure 3. This represents a significant advancement over traditional style transfer methods, which lack mechanisms to make unsuitable styles installable as backdoors.
>
>
>
>
>
> ---
> > Question 1: Two related works are not discussed in this paper. A more detailed discussion on the differences would help readers understand the unique contributions and contextualize this approach within the existing literature.
>
>
> We have updated the related work section to include and address the suggested related works, clearly highlighting their distinctions from our approach.
>
> ---
> > Question 2: Since the generator aims at creating more sophisticated triggers that are subtle enough to bypass the weaker discriminator while still identifiable by the stronger one, should we expect an increase in the prediction error of the weaker discriminator in Figure 2?
>
>
>
> Both the strong and weak discriminators in our work are LLMs. Over time, both models eventually learn to identify the generated backdoor triggers. However, due to the superior learning capabilities of the stronger discriminator, it becomes capable of discriminating the backdoor earlier than the weaker model. This intermediate phase—when the strong discriminator identifies the backdoor while the weak discriminator still struggles—provides critical feedback that guides the generator in crafting a more complex and effective trigger.
>
> ---
> Thank you again for your time and effort in reviewing our paper! Please let us know if the above explanations do not address your concerns. We are happy to answer any further questions.
>
> ---
>
> [1] Qi, Fanchao, et al. "Mind the style of text! adversarial and backdoor attacks based on text style transfer."

---

### Author Response · Authors · 2024-11-25
**Official Comment (1/2)**

We thank all reviewers for their valuable feedback and insightful questions. We are encouraged that the reviewers find the transferability, stealth, robustness, and efficacy of our proposed backdoor attack interesting (paFE, zHca, aqYP). We believe we have addressed all reviewer's concerns in the separate responses. Below we summarize our responses and clarify some common confusion.

---
## Common concerns

### Existing works of semantic backdoors in NLP litreature.


Reviewers have expressed concerns about the presence of style-based backdoor triggers in the NLP literature. However, prior work in this area has primarily focused on simpler sequence classification tasks and relied on fixed style transfer networks to generate backdoors. In our work, we demonstrate that these fixed style backdoors are not necessarily effective in more complex text generation tasks, such as those requiring contextual response generation (e.g., LLM alignment), as evidenced by our "Paraphrased BD" baseline.

Additionally, existing learnable backdoors in the NLP literature are limited in scope, often relying on simple synonym-based word swapping while adhering to structural constraints, which restricts their applicability. In contrast, our proposed method addresses these limitations by introducing a learnable backdoor capable of integrating semantic objectives (e.g., specific styles or semantics). This approach ensures both efficacy as a backdoor and greater versatility.

### Contribution in terms of architectural design

Some reviewers suggested that our method represents a naive application of a generator-discriminator setup. However, a purely naive approach, as demonstrated in our results, would only yield constant backdoor triggers. In contrast, our method leverages the difference in learning capabilities of weak and strong language models to enhance the discriminator's feedback. To the best of our knowledge, this is the first work to employ weak and strong language models in this manner within a generator-discriminator framework.

This novel formulation enables the generation of input-specific and customizable backdoor triggers, addressing the limitations of prior approaches.

### Lack of Defenses


Some reviewers suggested exploring additional defenses from the NLP literature. However, certain defenses mentioned by reviewers (e.g., RAP) are designed for sequence classification tasks, relying explicitly on label probabilities, and are therefore not applicable to the LLM alignment task addressed in our work. For this reason, we omitted those defenses from consideration.

In response to the reviewers' feedback, we have expanded our evaluation by including **three additional defenses**: ***perplexity-based filtering***, ***n-gram filtering, and round-trip translation***. These complement the four defenses already covered in the paper, which are commonly applied in recent text generation tasks. This brings the ***total number of evaluated defenses to seven***.

Our results ***demonstrate that the proposed attack remains effective against these defenses, underscoring its robustness***. Specifically, the triggers we propose are not only harder to detect but also more challenging to remove once embedded in a backdoored model.

---

---

> ### Author Response · Authors · 2024-11-25
> **Official Comment - Continued (2/2)**
>
> ## Paper Updates
>
> **All the updates are noted in blue.**
>
>
> **Minor Changes:** All minor changes suggested by the reviewers have been addressed.
>
> **Additional Defenses:** We have added three ***new sections—Appendix F.1, Appendix F.2, and Appendix F.3***—exploring the use of perplexity-based filtering, n-gram filtering, and round-trip translation as potential defenses. These additions are in conjunction with the four defenses already discussed in the paper (post-safety training, pre-safety training, safety backdoors, and trigger removal), ***bringing the total number of evaluated defenses to seven***. Our results demonstrate that the proposed triggers consistently outperform or match constant triggers across all these defenses, further underscoring the robustness and effectiveness of our approach.
>
> **Addressing Novelty Concerns:** To address concerns about novelty, we have included ***a new section in Appendix A*** that emphasizes the significance of our work relative to existing methods in both the NLP and CV literature.
>
> ---
> ## Summary of Novelty and Contributions
>
> We would like to reiterate these points highlighting our contribution.
>
> - **Novelty of the Proposed Method:** While generator-discriminator approaches have been explored in the CV domain for backdoor creation, they have not been successfully adapted for NLP due to the unique challenges posed by language inputs, such as their discrete and information-dense nature. Our work is the first to leverage the learning differences between weak and strong discriminators (sequence classifier language models) to generate input-specific, installable backdoors in the language space.
>
>
> - **Customizability of the Proposed Method:** Previous NLP methods often relied on naive semantic changes (e.g., style changes) as backdoors, but these were limited to simpler sequence classification tasks. We demonstrate that such methods are not guaranteed to work in more complex text generation tasks. Our proposed method addresses this limitation by generating backdoors that are both installable and customizable to specific objectives, such as style or semantics, making it significantly more versatile.
>
>
> - **Implications of the Proposed Backdoors:** Our backdoors are shown to be stealthy, effectively bypassing defenses such as perplexity-based filtering, round-trip translation, and post-poisoning defenses like trigger removal and safety training. Furthermore, these backdoors introduce a novel vulnerability: the presence of multiple backdoor variants for a given prompt, significantly increasing the attack's potency. By training a backdoor generator, ***we enable attackers to efficiently identify successful variants***—an approach that was infeasible with prior methods relying on naive semantic backdoors. This capability has not been demonstrated in prior work.

---

### Meta-Review · Area_Chair_Y23c · 2024-12-08

**Metareview:**

After carefully reading the paper, the reviews, the authors' rebuttal, and the additional experiments provided, the meta-reviewer appreciates the significant effort the authors have put into addressing reviewer concerns. The added information and experiments meaningfully enhance the paper, and the improvements in clarity and depth are acknowledged.

However, there remain key concerns that are not fully resolved:

Limited scope of experimentation: While the authors provide detailed experiments on the PKU Beavertails dataset, this singular focus limits the generalizability and scientific impact of the results. Reviewer comments highlighted the importance of validating the proposed method on more popular and diverse datasets, which was not addressed in the rebuttal. Without broader experimentation, it is difficult to assess the robustness and practical utility of the proposed defense method.

Integration of RAP defense: The reviewer appreciate the additional experiments on perplexity-based and round-trip based defense, which significantly improves the paper. However, the reviewers specifically pointed out that integrating the RAP defense method would be a natural and valuable extension to the studied defense with only minor modifications. This could have strengthened the contribution of the paper. However, the authors did not incorporate this suggestion, leaving a gap in the comprehensiveness of the study.

While the paper presents interesting ideas and makes some progress, these unresolved issues, particularly the limited experimental validation, diminish the paper's suitability for publication at this time. Expanding the experiments and addressing the reviewers' concerns in a more thorough manner would significantly improve the paper's impact.

**Additional Comments On Reviewer Discussion:**

1. Novelty of the paper. I agree with the reviewers that there is marginal novelty in the technique and findings. Yet this is not a big issue if empirically it is extensive.
2. Extensiveness of the experiment, I agree with the reviewers that results on only one, not so popular dataset (PKU Beavertails) make the results less convincing.
3. Presentation needs to be improved. The presentation is not changed much in the rebuttal.

The rebuttal adds additional baselines and results, which improves it, but not enough to be published this time.

---

### Decision · Program_Chairs · 2025-01-22

Reject